# Scaphoid Fracture Detection by Using Convolutional Neural Network

**DOI:** 10.3390/diagnostics12040895

**Published:** 2022-04-04

**Authors:** Tai-Hua Yang, Ming-Huwi Horng, Rong-Shiang Li, Yung-Nien Sun

**Affiliations:** 1Department of Biomedical Engineering, National Cheng Kung University, Tainan 701, Taiwan; dd2006tw@gmail.com; 2Department of Orthopedic Surgery, College of Medicine, National Cheng Kung University Hospital, National Cheng Kung University, Tainan 704, Taiwan; 3Department of Computer Science and Information Engineering, National Pingtung University, Pingtung 912, Taiwan; horng@mail.nptu.edu.tw; 4Department of Computer Science and Information Engineering, National Cheng Kung University, Tainan 701, Taiwan; geniuscat1227@gmail.com

**Keywords:** convolutional neural network, convolutional block attention module, faster R-CNN, feature pyramid network, scaphoid fractures

## Abstract

Scaphoid fractures frequently appear in injury radiograph, but approximately 20% are occult. While there are few studies in the fracture detection of X-ray scaphoid images, their effectiveness is insignificant in detecting the scaphoid fractures. Traditional image processing technology had been applied to segment interesting areas of X-ray images, but it always suffered from the requirements of manual intervention and a large amount of computational time. To date, the models of convolutional neural networks have been widely applied to medical image recognition; thus, this study proposed a two-stage convolutional neural network to detect scaphoid fractures. In the first stage, the scaphoid bone is separated from the X-ray image using the Faster R-CNN network. The second stage uses the ResNet model as the backbone for feature extraction, and uses the feature pyramid network and the convolutional block attention module to develop the detection and classification models for scaphoid fractures. Various metrics such as recall, precision, sensitivity, specificity, accuracy, and the area under the receiver operating characteristic curve (AUC) are used to evaluate our proposed method’s performance. The scaphoid bone detection achieved an accuracy of 99.70%. The results of scaphoid fracture detection with the rotational bounding box revealed a recall of 0.789, precision of 0.894, accuracy of 0.853, sensitivity of 0.789, specificity of 0.90, and AUC of 0.920. The resulting scaphoid fracture classification had the following performances: recall of 0.735, precision of 0.898, accuracy of 0.829, sensitivity of 0.735, specificity of 0.920, and AUC of 0.917. According to the experimental results, we found that the proposed method can provide effective references for measuring scaphoid fractures. It has a high potential to consider the solution of detection of scaphoid fractures. In the future, the integration of images of the anterior–posterior and lateral views of each participant to develop more powerful convolutional neural networks for fracture detection by X-ray radiograph is probably important to research.

## 1. Introduction

The scaphoid is the largest carpal bone in the human wrist and is close to the carpals and radius, see Figure 1 [1]. Its unique location and shape are prone to fracture as people fall and their palms strike any hard surface. The standard treatment of scaphoid is screw bone surgery because of its good recovery and short treatment time. However, the screw bone surgery needs precise positioning of the scaphoid and its fracture to plan an appropriate angle to implant screws. The small size of the scaphoid and the complex structures of carpals cause accurate screw implantation to be difficult, and thus it is a challenge.

A scaphoid fracture is usually diagnosed by an X-ray of the wrist [2]. However, a break in the bone that cannot yet be seen on X-ray is called an “occult” fracture. The occult fractures may be difficult to detect by eye observations because their occurrence probability has been estimated at 7–21% reported in a recent prospective study [3]. If pain persists, a follow-up exam and X-ray in a week or two can be used to diagnose it. Sometimes, a CT scan or MRI is used to obtain better views of the shape and alignment of the scaphoid and assist with the diagnosis or surgery plans, but it is very expensive. Therefore, it is critical to develop a more precise X-ray diagnosis of scaphoid fractures.

Image processing technologies are widely used to separate the region segmentation methodology of X-ray images, but they always need manual intervention to decide the boundary of scaphoid fractures. To date, the convolutional neural network (CNN) has advanced development worldwide, successfully being applied to several areas of medical diagnosis and robotics [4,5,6]. Langerhuized et al. [7] used CNN to detect scaphoid fractures on the conventional radiographs. In this paper, two consecutive CNNs are developed. One is used for scaphoid segmentation, while another is for fracture detection. The segmentation CNN localized the scaphoid and then passed them into the detection CNN for fracture detection. The resulting performance is the Area Under the Curve (AUC) of Receiver Operating Characteristic (ROC) of 0.77, 72% accuracy, 0.84 sensitivity, and 0.6 specificity.

Yoon et al. [8] isolated the scaphoid area in a bounding box by using the cascade RCNN model [9] and then fed them to the EfficientNetB3 neural network [10] to determine whether the scaphoid was fractured. The results of the cascade R-CNN model achieved an overall sensitivity and specificity of 87.1% and 92.1% (AUC = 0.995). The second EfficientNetB3 obtained an overall sensitivity of 79.0% and specificity of 71.6% with an AUC of 0.810.

Hendrix et al. [11] proposed two consecutive CNNs including a segmentation CNN for scaphoid segmentation and a detection CNN for detecting the fractures. The segmentation CNN localized and then cropped scaphoid area. The cropping area was resized into a fixed size and normalized its contrast. The detection CNN was based on the DenseNet-121 model. A class activation map was then calculated using the smooth Grad-CAM++ algorithm [12]. The resulting segmentation CNN achieved a Dice of Coefficient (DSC) of 97.4% and symmetric Hausdroff Distance (HD) of 1.31 mm. The detection CNN had an overall sensitivity of 78.0% and specificity of 84% with an AUC of 0.87.

Tung [13] also used two CNNs to detect scaphoid fractures using the YOLO-v4 CNN model for scaphoid area detection and classification CNN to determine whether the detected scaphoid has fractured. The different backbones such as the VGG, ResNet [14], DenseNet [15], InceptionNet [16], and EfficientNet used to construct classification CNN. The experimental results showed that DenseNet 201 and ResNet 101 are more promising selections. The reported performances of the DenseNet 201 backbone are sensitivity of 0.833, specificity of 0.611, precision of 0.682, F1-score of 0.750, AUC of 0.444, and accuracy of 0.722.

Even though many articles have attempted to assess the detection issues of fractures in the X-ray images, the detection results of those articles do not meet the requirement of clinical diagnosis. The paper proposes a two-stage fracture detection of scaphoid images based on the convolutional neural networks. The contribution of this study can be described as follows:
To increase detection accuracy, the proposed method consists of two CNNs: one is to identify the scaphoid area, and another is to detect the fractures of scaphoid.Identifying scaphoid areas can reduce the searching space and computational times for consequent fracture detection.A powerful fracture detection CNN consists of ResNet, spatial feature pyramid (SSP), and convolutional block attention module (CBAM). Experimental results showed that the proposed CNN achieved high detection performance.The rest of this paper is organized as follows. The material data set and methods are proposed in detail in Section 2. Section 3 then presents the experimental results and empirical discussion. Finally, Section 4 provides the conclusions of this study.

## 2. Material and Method

This section describes the data collections, data augmentations, scaphoid bone detection, and fracture detection.

### 2.1. Data Collections and Implementation Environment

In this study, all experimental X-ray images were collected from the National Cheng Kung University Hospital (NCKUH) in Taiwan, including 280 adult patients. Specifically, 178 fracture instances of surgical verification were selected as the positive images, and 102 normal instances were considered the negative images. All programs were implemented in a personal computer with an i9 processing unit, 16 GB of main memory and a GeForce RTX 3090 GPU using the Windows 10 operating system. The performance of our proposed system is measured by 5-fold cross-validation method.

### 2.2. Methods

The proposed CNNs include the scaphoid detection CNN and the fractures detection CNN. The first one is based on the original Faster R-CNN model [17], and the other fractures detection is based on the rotation-decoupled anchors [18] to finely detect the fracture areas shown in Figure 2. The two CNNs are described as follows.

#### 2.2.1. Scaphoid Area Detection

The traditional CNN uses a sliding kernel to convolute the whole X-ray image to generate the feature maps for further object detection or segmentation. Yet, the scaphoid area must be limited to a small area of an X-ray image. In this paper, the scaphoid area detection used is the Faster R-CNN, which is faster and more precise than Fast R-CNN [19,20]. The Faster R-CNN consists of three components: feature map convolution network, region proposal network (RPN), and Fast R-CNN as shown in Figure 3.

The feature map convolution network was implemented by the ResNet 50 that consists of five stages. Each stage includes max pooling, the residual block, and convolutional + batch normalization + ReLU block were applied as the backbone for generating feature maps. The original X-ray image is resized into 1600 × 1200 pixels and the last output feature map was 75 × 100 with 1024 channels. In addition, the RPN and the Fast R-CNN shared this feature map. Typically, the RPN referred to box regressions and confidence scores of searching objects to generate several bounding boxes with different sizes. It then predicted objects by these bounding boxes and integrated them into some proposals. The corresponding region of interest (ROI) on the feature maps of these proposals was transferred to the Fast R-CNN for further use. The Fast R-CNN received proposals from the RPN with the corresponding ROI features from the shared feature maps. Different sizes of ROI features were max-pooled to a 7 × 7 feature map. The fixed-size feature map was fed into a sequence of fully connected layers and then connected into two sibling layers for classification and bounding box regression. The classification gave the detection confidence score, and the regression gave the position regression of the bounding box. The used loss function includes the classification loss and the regression loss in the training stage of the Faster R-CNN model.

#### 2.2.2. Fracture Area Detection

In this paper, the fractures detection CNN consists of the ResNet 152 backbone, the feature pyramid network [21,22] (FPN, for generating the multiscale feature maps) and the convolutional block attention module [23] (CBAM; for determining whether it is fractured). The last three blocks comprise the feature pyramid network. The number of layers used by the FPN is three rather than the original four because of better performances in experiments. The resulting feature maps of FPN are fed into the predict head for detecting the fracture areas. The predict head has two branches: the *cls* branch (detecting positive/negative bounding box) and the *loc* branch (regressing the position of fracture bounding box). The two branches are implemented in Conv + BN + ReLU + Conv, and the shapes of the output of cls are [B, A × C] and loc is [B, A × 5] (where B, A, C are denoted to the numbers of the batch, anchor and class).

The detailed structure of FPN is shown in Figure 4. The FPN passes the deep features layer by layer and directly combines or concatenates the deep and shallow features to generate high-resolution feature maps and desirable semantics.

The CBAM enhances the input feature *F* of FPN into refined feature $F^{\prime }$ by inferring the one-dimensional (1D) channel attention map $N_{c}\in R^{1\times 1\times C}$, and the 2D spatial attention map $N_{s}\in R^{H\times W\times 1}$ as shown in Figure 5 [24,25]. The entire attention process can be summarized as follows:(1)$$F^{\prime }=N_{s}(N_{c}\left(F\right)☉F)☉N_{c}\left(F\right)♁F$$
where ☉ and ♁ denote the elementwise product and sum. The following describes the two attention modules as shown in Figure 6.

Channel attention module. The aggregated feature map is combined through average-pooling and max-pooling operations. The two resulting feature maps are represented as $F_{ave}^{c}$ and $F_{max}^{c}$ with the sizes of 1×1×c. These two feature maps are independently built from two fully connected layers and then integrate them into a feature map $N_{c}\left(F\right)$ by elementwise addition and sigmoid operations.Spatial attention module. The channel attention feature map $N_{c}\left(F\right)$ and original feature map *F* are aggregated into the new feature map, and down-sampled by using the average-pooling and max-pooling operation. The results generate two different feature maps $F_{ave}^{s}$ and $F_{max}^{s}$, with sizes of *H* × *W* × 1. The two feature maps are concatenated into larger maps and then convoluted and activated by a 7 × 7 × 2 kernel and sigmoid function.

The image-wise classifier is established with a full connected network block. The binary classification cross-entropy loss used is defined as follows.
(2)$$L_{image}BCE\left(y,p\right)=\{\begin{array}{cc} -\log\left(p\right) & if\;y=1\\ -\log\left(1-p\right) & if\;y\neq 1 \end{array}$$
where *y* represents a scaphoid image and y = 1 means the scaphoid image y has fractured, and the y≠1 means no fracture occurs.

A rotation-decoupled detector (RDD) of the predicted head of this paper is used to detect oriented bounding box (OBB) with additional rotation angle because the fractured area of scaphoid always emerges with arbitrary oriented angles, dense distribution and a larger aspect ratio. The horizontal bounding box (HBB) is usually represented by (x,y,w,h), where (*x*, *y*) is the center, and *w* and *h* are the lengths of the X and Y axes. The OBB is represented by (*x*, *y, w*, *h*, θ) where θ is the angle of the bounding box. To combine the advantages of HBB and OBB, the detecting rotational bounding box is redefined as the horizontal bound box HBBhv and an angle θ ranged in [−π4,π4], where the detected OBB has the same center, width, and length as the HBBhv horizontal bound box. To accelerate the training and interfacing process, only horizontal anchors are used. In the anchor matching, the rotational ground truth box is decoupled to a HBBhvT and an acute angle $\theta_{T}$ for matching. The computation of intersection of union (IOU) between OBB and ground truth is disregard of their angles, as shown in Figure 7 [25].

After matching, most of the bound boxes are classified as positive objects or negative backgrounds based on the IOU thresholds set to 0.5 and 0.4. In other words, if the IOU of the bounding box is more than 0.5 or less than 0.4, it will be assigned the positive or negative case, and the remaining one is considered the ignored bounding box. In general, the number of negative bounding boxes is much larger than the positive case. The positive-negative samples imbalance problem always results in inaccurate classification. The popular balance strategy is Focal Loss [26]; the Focal Loss method assigned the categorical labels of positive, negative, and ignored bounding to 1, 0, and −1. The corresponding binary cross-entropy (BCE) is defined as follows:(3)$$BCE\left(y,p\right)=\{\begin{array}{cc} \log\left(p\right) & if\;y=1\\ -\log\left(1-p\right) & if\;y=0\\ 0 & if\;y=-1 \end{array}$$
where y is the class label of a bounding box and p is its predicted probability.

The classification Loss $L_{cls}$ is defined as follows.
(4)$$L_{cls}=\frac{1}{\#\left(postive\;bboxes\right)}\sum_{i=1}^{N}BCE\left(y_{i},p_{i}\right)$$
where the *N* and the #(*positive boxes*) denote the number of all bounding boxes and the number of positive bounding box.

The ground truth box and prediction box are represented as *v* = ($t_{x},\;t_{y},\;t_{w},\;t_{h},t_{\theta }$) and $v^{*}=\left(t_{x}^{*},t_{y}^{*},\;t_{w}^{*},\;t_{h}^{*},\;t_{\theta }^{*}\right)$ for position regression. If a corresponding anchor is expressed as $\left(x_{a},y_{a},w_{a},h_{a}\right)$, the $v^{*}$ is defined in the following equations:(5)tx=x−xa/wa, ty=y−ya/hatw=log(w/wa), th=log(h/ha)     tθ=4θπtx*=x*−xa/wa, ty*=y*−ya/hatw*=log(w*/wa), th*=log(h*/ha)      tθ*=tanh(θ*)

The $smooth_{L1}$ loss for the rotation bounding box regression is used for the object anchors.
(6)$$L_{reg}=\frac{1}{\#\left(postive\;bboxes\right)}\sum_{i=1}^{\#\left(postive\;bboxes\right)}smooth_{L1}\left(v_{i}^{*}-v_{i}\right)$$
where,
(7)$$smooth_{L1}\left(x\right)=\{\begin{array}{cc} 0.5x^{2} & if\;\left|x\right|<1\\ \left|x\right|-0.5 & otherwise \end{array}$$

Finally, the multi-task loss is defined as:(8)$$L=L_{cls}+\alpha L_{reg}+\frac{1}{\beta }L_{image}BCE\left(y,\;p\right)$$
where β is the batch size.

#### 2.2.3. Performance Evaluation

In this section, the performance metrics of the proposed system were evaluated [27,28,29,30]. There are four performance metrics, including the *Accuracy*, *Recall*, *Precision*, and F *scores* are defined as:(9)$$Accuarcy\left(AC\right)=\frac{TP+TN}{TP+TN+FP+FN}$$
(10)$$Sensitivity\left(SE\right)=\frac{TP}{TP+FN}$$
(11)$$Specificity\left(SP\right)=\frac{TN}{TN+FP}$$
(12)$$Recall\left(R\right)=\frac{TP}{TP+FN}$$
(13)$$Precision=\frac{TP}{TP+FP}$$
(14)$$F-scores=\frac{2\times Precision\times Recall}{Precision+Recall}$$

The *TP*, *TN*, *FP*, and *FN* denote the number of true positive, true negative, false positive, and false negative.

In addition, the Receiver Operating Characteristic (ROC) curve [31,32] is a graphical plot that illustrates the capability of a binary classification system. The ROC curve is created by plotting the true positive rate against the false-positive rate at different thresholds. The Area Under the Curve (AUC) is a powerful measure of the ability of a classifier to distinguish between classes. The AUC measures the ability of a classifier to distinguish classes and is used as a summary of the ROC curve. The AUC ranges from 0 to 1. The higher the AUC, the better the performance of the model at distinguishing between the positive and negative classes.

## 3. Results

This section describes the experimental results of the scaphoid and fracture detection.

### 3.1. Results of Scaphoid Detection

A total of 361 X-ray scaphoid radiographs were used including the 167 fractured samples and 194 normal samples. Approximately, there were 18.56% (i.e., 31/167) occult fractured samples in the 167 fractured samples. Figure 8 shows the two examples of the X-ray images with fracture and without fracture. All sample images were verified using the 5-fold cross-validation method. Three different data augmentation methods were used; (1) contrast limited adaptive histogram equalization (CLAHE) [33], (2) random horizontal flip with 50% probability, and (3) random contrast with 50% probability. The used CNN is the Faster R-CNN model with a backbone of ResNet 50. The trained strategy is the learning rate of 0.001, optimizer is SGD, batch size of 1 and epochs of 10,000. In total, the number of parameters of the CNN model of scaphoid detection is 41,755,286 and the parameters are pre-trained by using the PASCAL-VOC [34]. The training time of CNN is about 61 min and the prediction time of each test image is 0.267 s. The classification accuracy tested in the 361 samples based on the five times of 5-fold cross-validations is 0.997 in which only one fracture scaphoid image failed to detect. Similar result of classification accuracy of 0.994 was also found in Tung’s work [13].

### 3.2. Fracture Detection and Classification

The detected scaphoid images also include the 166 fractured samples (previous one scaphoid image is discarded) and 194 normal samples. Figure 9 shows the two examples of the detected scaphoid images with/without fractures. The data augmentation and 5-fold cross-validation methods are used the same as the scaphoid detection. The trained strategy is the learning rate of 0.001, optimizer is SGD, batch size of 1, and epochs of 10,000. As shown in Table 1 and Table 2, we compute the averages of recall, precision, sensitivity, specificity, accuracy, and AUC using the five times of 5-fold cross-validations methods. In total, the number of parameters of the CNN model of fracture detection is 66,913,876 and the parameters are pre-trained by using the PASCAL-VOC [34]. The training time of CNN is about 37 min and prediction time of each test image is 0.028 s.

Compared to other works of fracture detections, Yoon [6] presented the sensitivity of 0.79, the specificity of 0.716, and the AUC of 0.81. From Table 1, the sensitivity of 0.789, the specificity of 0.900, and the AUC of 0.92 are found; these performance measures are superior to the results of Yoon, especially as the specificity of our proposed method markedly precedes the work of Yoon.

The works of the fracture classifications in the literature showed that Tung [13] presented a sensitivity of 0.833, specificity of 0.611, and AUC of 0.79. Yoon [8] presented a sensitivity of 0.87, specificity of 0.92, and AUC of 0.96. Langerhuizen [7] presented a sensitivity of 0.84, specificity of 0.6, and AUC of 0.81. Hendrix [11] presented a sensitivity of 0.78, specificity of 0.84, and AUC of 0.87. While our result is worse than the work of Yoon, it is superior to other three methods.

However, the recalls of our proposed results for the fracture detection and classification are only 0.789 and 0.735. The results mean that a higher false-negative rate occurs, meaning there may exist a large number of misclassifications of positive images with occult fractures. To understand the results of the 31 occult fracture samples, their detection and classification results are shown in Table 3. The results showed that the correct detection and classification is only 50%.

## 4. Conclusions

This study delivers evidence supporting the CNN approaches that accurately detect the fracture areas and classify the status of scaphoid fractures using X-ray scaphoid radiographs. The proposed model includes two consecutive CNNs to detect the scaphoid and classify scaphoid fractures. Experimental results reveal that the detection of the scaphoid area performs well, but the recalls of the detection and classification of scaphoid fractures are only 0.789 and 0.735. It therefore implies high false-negative rates and warrants further exploring the results of 31 occult samples. We found that only 50% correct fracture classification and detection were reached in testing of the 31 occult samples.

In the future, the improvement of the classification capability of occult samples is still an important research topic. The integration of the anterior-poster and lateral view images of each participant is a probable approach to developing more powerful convolutional neural networks for fracture detection with X-ray radiographs.

## Figures and Tables

**Figure 1 diagnostics-12-00895-f001:**
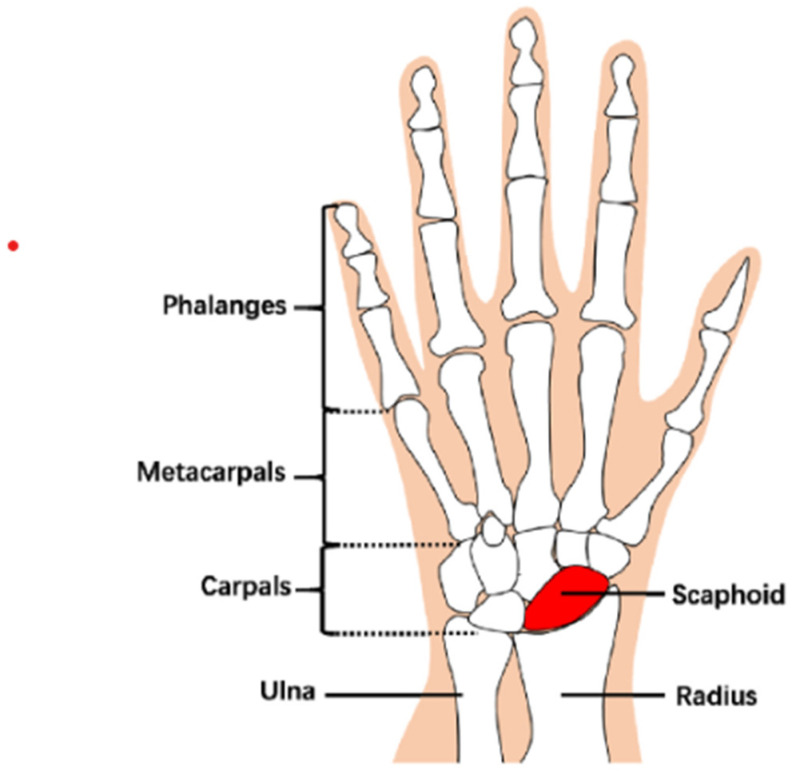
The anatomical structures of the wrist bone, in which the region with red is the scaphoid.

**Figure 2 diagnostics-12-00895-f002:**
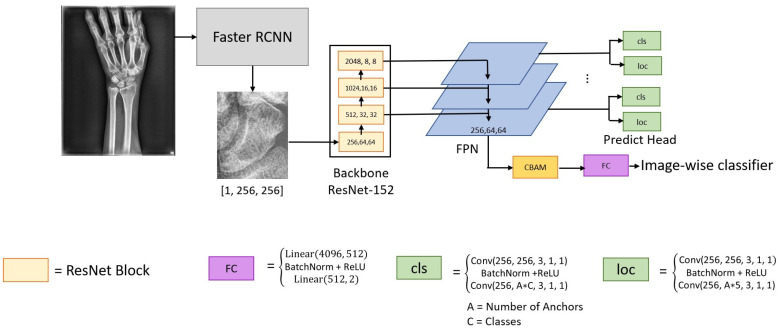
The CNNs used include scaphoid detection and fracture detection.

**Figure 3 diagnostics-12-00895-f003:**
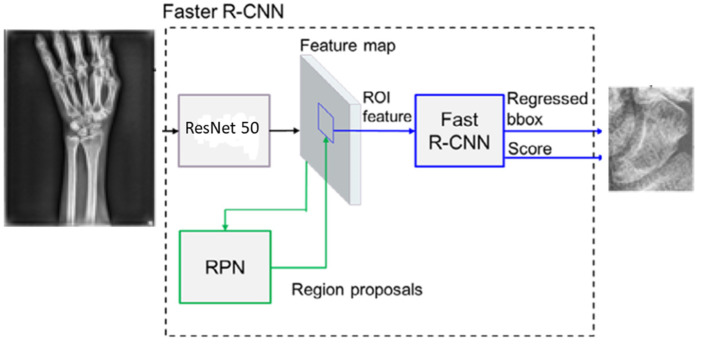
The structure of the Faster R-CNN model.

**Figure 4 diagnostics-12-00895-f004:**
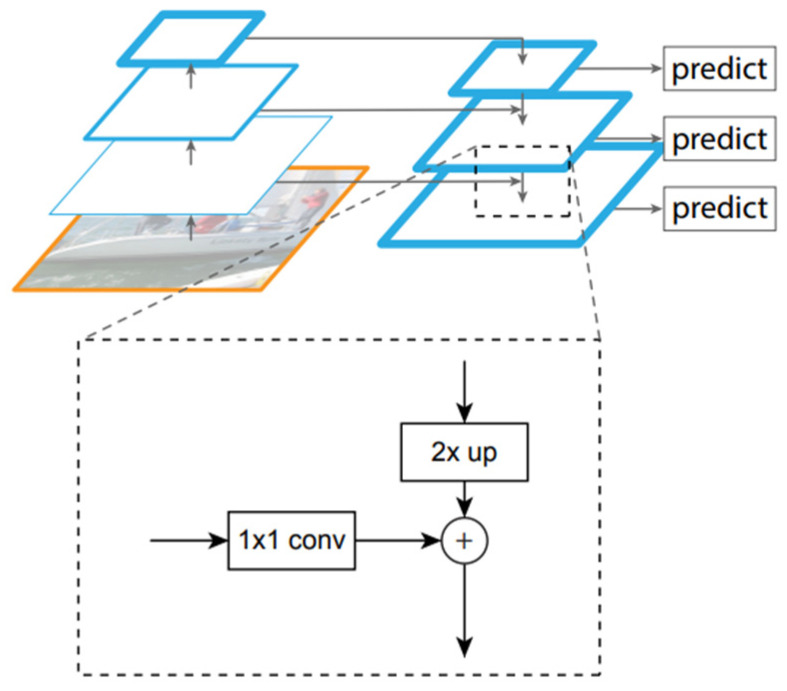
The structure of the feature pyramid network; ♁ is the elementwise sum.

**Figure 5 diagnostics-12-00895-f005:**
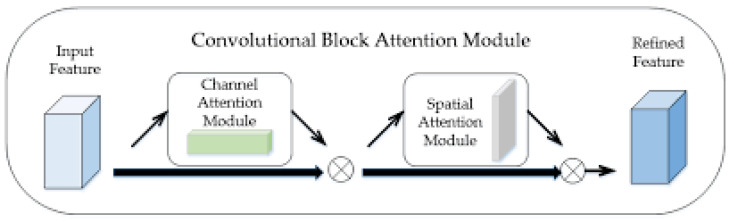
The structure of the convolutional clock attention module.

**Figure 6 diagnostics-12-00895-f006:**
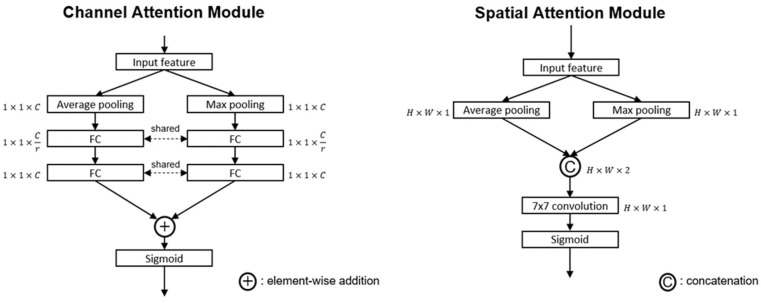
The structure of the attention module, *H*, *W*, and *C* are the height, width, and channel of feature map.

**Figure 7 diagnostics-12-00895-f007:**
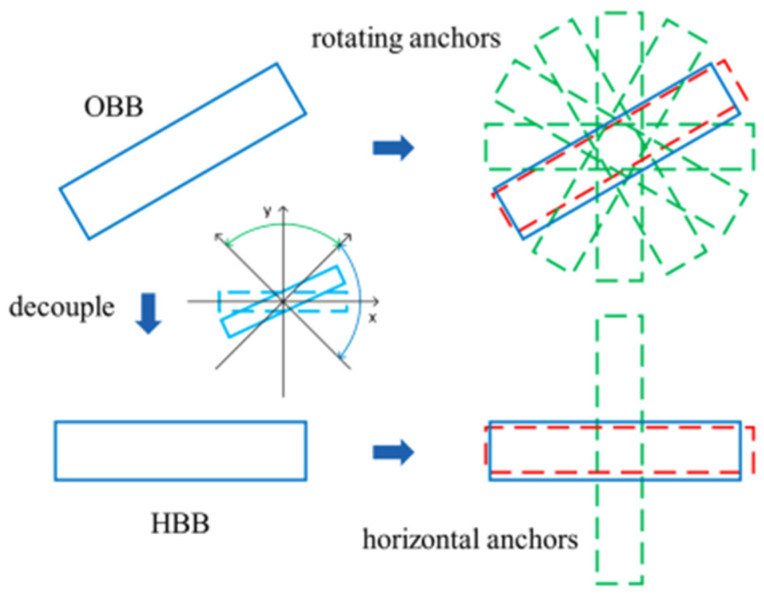
Matching methods of the ground truth bounding box and the rotational bounding box.

**Figure 8 diagnostics-12-00895-f008:**
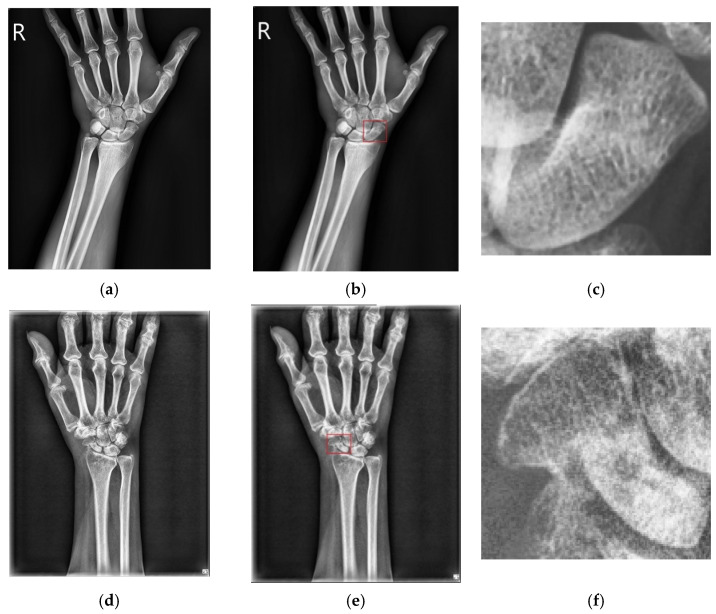
Two image samples with and without fractures. (**a**) Normal case. (**b**) Scaphoid area marked red of (**a**). (**c**) Enlarged scaphoid area of (**b**). (**d**) Fractured case. (**e**) Scaphoid area of (**a**). (**f**) Enlarged scaphoid area of (**b**).

**Figure 9 diagnostics-12-00895-f009:**
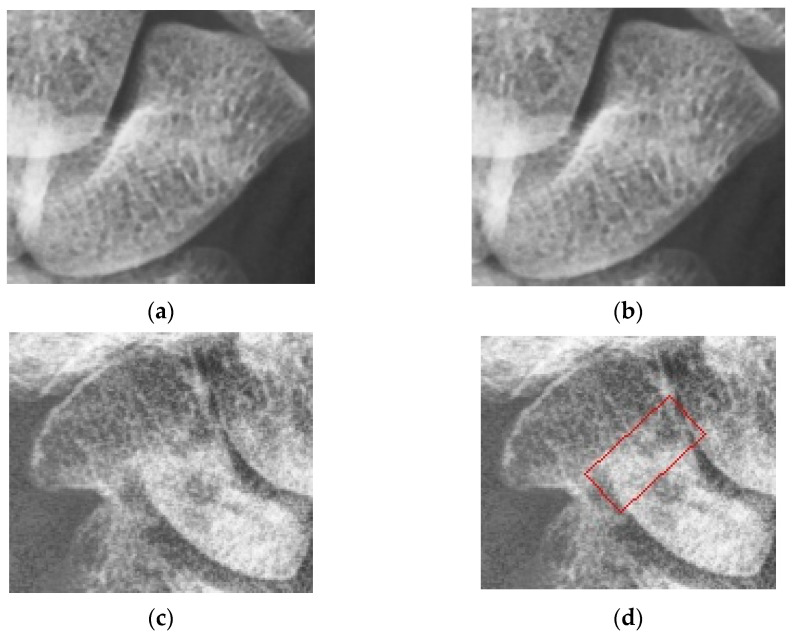
Two scaphoid cases with/without fractures. (**a**) Scaphoid case without fracture. (**b**) No fracture bounding box. (**c**) Scaphoid case with fracture. (**d**) Fracture bounding box.

**Table 1 diagnostics-12-00895-t001:** The results of fracture detection of detected scaphoid.

Methods	Recall	Precision	Accuracy	Sensitivity	Specificity	AUC
Our proposed method	0.789	0.894	0.853	0.789	0.900	0.920

**Table 2 diagnostics-12-00895-t002:** The results fracture classification of detected scaphoid.

Methods	Recall	Precision	Accuracy	Sensitivity	Specificity	AUC
Our proposed method	0.735	0.898	0.829	0.735	0.920	0.917

**Table 3 diagnostics-12-00895-t003:** The results of fracture detection and classification of 31 occult samples.

Methods	Accurate Samples	Inaccurate Sample
Detection	15	16
Classification	16	15

## Data Availability

Not applicable.

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
