# Peer review of "Scaphoid Fracture Detection by Using Convolutional Neural Network"

_diagnostics, 2022, doi:10.3390/diagnostics12040895_

Round 1

Reviewer 1 Report

Paper is important for health informatics. It needs some essential corrections before publication-

  • The full form of ROC must be ‘receiver operating characteristic.’ It is generally known as ROC curve.
  • How are selection of layers done in CNN? Author must explain.
  • The conclusion of the paper must be revised. Author should explain future scope of the work.
  • Authors must check typo errors.
  • What are the main noises which are encountered in the proposed work?
  • Add some important findings in the abstract by explaining lab setting and novelty of the work.
  • Revise the literature survey by adding demerits of previous techniques and merits of proposed techniques.
  • All equations should be clearly explained by adding explanation on all associated variables.
  • How to choose parameters for simulation? Give proper explanation.
  • Highlight the more applications of proposed technique for variety of applications. Is it important for robotics applications?
  • What are the computational complexities involved in the proposed technique?
  • More explanation on results and tables are required.
  • What are the main limitations of proposed work? Author may refer below papers for ready reference.
  • Author should add flow charts of the proposed methodology.
  • The sequence of the sections should be- (1) Introduction, (2) Related Work, (3) Methodology, (4) Critical Outcomes, (5) Discussion, (6) Conclusion, (7) Future Scope.
  • Add following good quality papers which are based on different techniques, confusion matrix parameters, and merits/demerits of different techniques. After extensive review, suggesting these papers which are essential for improving quality of the paper-

-FrWT-PPCA-Based R-peak Detection for Improved Management of Healthcare System

-R-peak detection for improved analysis in health informatics

-An efficient ALO-based ensemble classification algorithm for medical big data processing

-Bio-medical analysis of breast cancer risk detection based on deep neural network

-BP Signal Analysis Using Emerging Techniques and its Validation Using ECG Signal

-R-peak detection based Chaos analysis of ECG signal

-A comparison of ECG signal pre-processing using FrFT, FrWT and IPCA for improved analysis

-A novel method of cardiac arrhythmia detection in electrocardiogram signal

-R-peak Detection Using Chaos Analysis in Standard and Real Time ECG Databases

-Chaos theory: An Emerging tool for Arrhythmia Detection

-Performance evaluation of various pre-processing techniques for R-peak detection in ECG signal

-Chaos theory and ARTFA: Emerging tools for interpreting ECG signals to diagnose cardiac arrhythmias

-A Critical Review of Feature Extraction Techniques for ECG Signal Analysis

-QRS Complex Detection Using STFT, Chaos Analysis, and PCA in Standard and Real-Time ECG Databases

Author Response

Dear Reviewer: 

All reply are listed in the attached file. Thanks. 

Comments

All revision are marked in red color.

1.The full form of ROC must be ‘receiver operating characteristic.’ It is generally known as ROC curve.

ROC had been revised into “receiver operating characteristic” in whole revised paper. .

2. How are selection of layers done in CNN? Author must explain.

(lines 157-159) The explanation cab be found in lines 157-159.

3. The conclusion of the paper must be revised. Author should explain future scope of the work.

(lines 336-339) The further research had been added to the conclusion

4. Authors must check typo errors.

Some typo errors had been revised.

5. What are the main noises which are encountered in the proposed work?

The fractures in the scaphoid bone that cannot be seen on x-ray yet is called an “occult” fracture. The occult fractures may be difficult to detect by eye observation because their occurrence probability about 7%-21%. The objective of this paper is to develop CNNs model to improve the detection of occult fractures. To date, there are not reliable guidelines to develop the CNN models for the detection of scaphoid fractures. 

6. Add some important findings in the abstract by explaining lab setting and novelty of the work.

(lines 14-38) The abstract had been revised.

7. Revise the literature survey by adding demerits of previous techniques and merits of proposed techniques.

Some good quality papers are referred in references 28-32.  

8. All equations should be clearly explained by adding explanation on all associated variables.

All variables are clearly explained.

9. How to choose parameters for simulation? Give proper explanation.

(lines 280-285 and lines 295-298)

All parameters had been explain in lines 280-283 and lines 295-298

10. Highlight the more applications of proposed technique for variety of applications. Is it important for robotics applications?

(lines 59-63) The variety application including the robotics and related reference had been added in lines 59-62.

11. What are the computational complexities involved in the proposed technique?

(lines 282-283 and lines 297-300)

The parameters of two CNNs, train time and predict time are added.

12. More explanation on results and tables are required.

(lines 304-308, lines 318-321)

Some explanations had been added in lines 304-308, lines 318-321.

13.

1. What are the main limitations of proposed work?

2.Author may refer below papers for ready reference.

1. (lines 59-63 of page 2). The image processing technologies are widely used to separate the region segmentation methodology of X-ray images, but it always need the manual intervention to decide the boundary of scaphoid fractures. To date, the convolutional neural network (CNN) has advanced development worldwide, successfully being applied to several areas of diagnosis of fractures on plain radiographs.

2. (reference 28) I had added the reference “Bio-medical analysis of breast cancer risk detection based on deep neural network” in the “Introduction” section.

14. Author should add flow charts of the proposed methodology.

The proposed method is an end-to-end mechanism. The X-ray image is sent to the CNN used as shown in Figure 2 without pre-processing stage. Therefore, the Figure 2 can be considered as the flow charts.   

15. The sequence of the sections should be- (1) Introduction, (2) Related Work, (3) Methodology, (4) Critical Outcomes, (5) Discussion, (6) Conclusion, (7) Future Scope.

According to the “Instructions for Authors” of the Diagnostics of MDPI Publisher, all manuscripts must contain the required sections: “Author Information, Abstract, Keywords, Introduction, Materials & Methods, Results, Conclusions, Figures and Tables with Captions, Funding Information, Author Contributions, Conflict of Interest and other Ethics Statements.”, Therefore, the proposed paper must follow up this requirement.

16. Add following good quality papers which are based on different techniques, confusion matrix parameters, and merits/demerits of different techniques. After extensive review, suggesting these papers which are essential for improving quality of the paper-

-FrWT-PPCA-Based R-peak Detection for Improved Management of Healthcare System

-R-peak detection for improved analysis in health informatics

-An efficient ALO-based ensemble classification algorithm for medical big data processing

-Bio-medical analysis of breast cancer risk detection based on deep neural network

-BP Signal Analysis Using Emerging Techniques and its Validation Using ECG Signal

-R-peak detection based Chaos analysis of ECG signal

-A comparison of ECG signal pre-processing using FrFT, FrWT and IPCA for improved analysis

-A novel method of cardiac arrhythmia detection in electrocardiogram signal

-R-peak Detection Using Chaos Analysis in Standard and Real Time ECG Databases

-Chaos theory: An Emerging tool for Arrhythmia Detection

-Performance evaluation of various pre-processing techniques for R-peak detection in ECG signal

-Chaos theory and ARTFA: Emerging tools for interpreting ECG signals to diagnose cardiac arrhythmias

-A Critical Review of Feature Extraction Techniques for ECG Signal Analysis

-QRS Complex Detection Using STFT, Chaos Analysis, and PCA in Standard and Real-Time ECG Databases

Some good quality papers are referred in references 28-32.

Reviewer 2 Report

The manuscript proposes and assesses a two-stage method to construct convolutional neural networks to detect scaphoid fractures from x-ray images.
Experimental results showed a scaphoid bone detection with an accuracy of 99.70%, fracture detection with an accuracy of 85.3%, fracture classification with an accuracy of 82.9%. 
I find the topic interesting and being worth of investigation and the document is well strucutred, organized, fluidly written, has enough background information, the methodology followed is clearly explained, the formulas used are correct, the results are clearly presented and support the conclusions.
Although I propose the following suggestions:
- I strongly suggest authors from refraining using personal pronouns such as "we" and "our" throughout the text and I encourage them to write it in an impersonal form of writing.
- Abstract requires structuring such as: problem, motivation, aim, methodology, main results, further impact of those results.
- keywords should be in alphabetical order.
- No further research is disclosed at the conclusion.

Author Response

Dear reviewer:

All reply had been completed and listed in the attached file. 

Thanks for your valuable comments.

Comments

All revisions are marked in red color.

Reviewer2

Reply

1. I strongly suggest authors from refraining using personal pronouns such as "we" and "our" throughout the text and I encourage them to write it in an impersonal form of writing.

The sentences using personal pronouns had been revised.

2. Abstract requires structuring such as: problem, motivation, aim, methodology, main results, further impact of those results.

(lines 14-38) The abstract had been revised according to problem, motivation, aim, methodology, main results, further impact of those results.

3. keywords should be in alphabetical order.

(lines 39-40)The keywords are listed in alphabetical order.

4. No further research is disclosed at the conclusion

(lines 336-339) The further research had been added to the conclusion.

Round 2

Reviewer 1 Report

Authors have done all recommended corrections. Now paper is acceptable in current form.